

# Automated mapping of *Portulacaria afra* canopies for restoration monitoring with convolutional neural networks and heterogeneous unmanned aerial vehicle imagery

Nicholas C. Galuszynski[1,*], Robbert Duker[1,*], Alastair J. Potts[1] and
Teja Kattenborn[2,3]

[1] Department of Botany, Nelson Mandela University, Gqeberha, South Africa
[2] Remote Sensing Centre for Earth System Research (RSC4Earth), Universität Leipzig, Leipzig, Germany
[3] German Centre for Integrative Biodiversity Research (iDiv), Halle-Jena-Leipzig, Leipzig, Germany
* These authors contributed equally to this work.

Corresponding author
Alastair J. Potts,
alastair.potts@mandela.ac.za

## ABSTRACT

Ecosystem restoration and reforestation often operate at large scales, whereas monitoring practices are usually limited to spatially restricted field measurements that are (i) time- and labour-intensive, and (ii) unable to accurately quantify restoration success over hundreds to thousands of hectares. Recent advances in remote sensing technologies paired with deep learning algorithms provide an unprecedented opportunity for monitoring changes in vegetation cover at spatial and temporal scales. Such data can feed directly into adaptive management practices and provide insights into restoration and regeneration dynamics. Here, we demonstrate that convolutional neural network (CNN) segmentation algorithms can accurately classify the canopy cover of *Portulacaria afra* Jacq. in imagery acquired using different models of unoccupied aerial vehicles (UAVs) and under variable light intensities. *Portulacaria afra* is the target species for the restoration of Albany Subtropical Thicket vegetation, endemic to South Africa, where canopy cover is challenging to measure due to the dense, tangled structure of this vegetation. The automated classification strategy presented here is widely transferable to restoration monitoring as its application does not require any knowledge of the CNN model or specialist training, and can be applied to imagery generated by a range of UAV models. This will reduce the sampling effort required to track restoration trajectories in space and time, contributing to more effective management of restoration sites, and promoting collaboration between scientists, practitioners and landowners.

## INTRODUCTION

With the United Nations "Decade on Ecosystem Restoration" underway, there is likely to be a global increase in the extent of restoration initiatives. These initiatives will require methods for accurately monitoring the trajectories of restoration efforts in an efficient and cost-effective manner (*de Almeida et al., 2020*; *Méndez-Toribio, Martínez-Garza & Ceccon, 2021*; *Murcia et al., 2016*). Additionally, these methods should be easily transferable, allowing non-experts to collect data at large spatial and temporal scales. Recent advances in remote sensing technologies and deep learning algorithms may provide the tools required for restoration practitioners (*Brodrick, Davies & Asner, 2019*; *Kattenborn et al., 2021*; *Zhu et al., 2017*). The work presented here demonstrates that using standard and low-cost "out-of-the-box" unoccupied aerial vehicles (UAVs) coupled with convolutional neural network (CNN) algorithms allows for the detection and quantification of *Portulacaria afra* Jacq. across experimental restoration plots established between 2007 and 2008. *Portulacaria afra* is regarded as an ecosystem engineer in the Albany Subtropical Thicket biome (*van Luijk et al., 2013*; *Wilman et al., 2014*) and is the target species for large-scale restoration initiatives (*Mills et al., 2015*; *Mills & Cowling, 2006*; *van der Vyver, Mills & Cowling, 2021*). It is estimated that up to 1.2 million hectares of the Thicket biome is degraded to some extent (*Lloyd, van den Berg & Palmer, 2002*), and requires some form of intervention to prevent the complete collapse of ecosystem functioning. With approximately 7,000 ha of plantings having been completed by 2017 (*Mills & Robson, 2017*), it is likely that the scale of thicket restoration will reach the tens of thousands of hectares in coming years due to the surge of investment into the carbon market and restoration initiatives across the world. Thus, the approach presented here could be invaluable for the cost-effective monitoring and management of thicket restoration initiatives.

The Thicket biome is largely confined to the Eastern Cape Province of South Africa and is characterized as a low-growing, spinescent, dense woodland system with high standing biomass often dominated by a matrix of the succulent shrub, *P. afra* (*Vlok, Euston-Brown & Cowling, 2003*). Occurring within a semi-arid environment, the high productivity of thicket is globally unique, with litter production rates comparable to that of some temperate forest systems (*Lechmere-Oertel et al., 2008*). This productivity has formed the foundation for the wool and mohair industry in the region (*Beinart, 2008*; *Oakes, 1973*; *Stuart-Hill, 1992*). While reported to be resistant to herbivory by certain indigenous browsers (*Stuart-Hill, 1992*, but see *Landman et al., 2012*), thicket vegetation can be rapidly cleared of *P. afra* (due to the species' high palatability) when subjected to intensive browsing by domestic livestock (*Hoffman & Cowling, 1990*; *Lechmere-Oertel et al., 2008*). This results in a structural shift from a dense, closed-canopy woodland to an open habitat consisting of a handful of remnant and isolated woody species that occur within a matrix of bare soil, ephemeral herbs, grasses and dwarf shrubs (*Lechmere-Oertel et al., 2008*; *Sigwela et al., 2009*; *Stuart-Hill, 1992*).

The change in *P. afra* cover due to unsustainable live-stock management pushes the system to a point where the natural regeneration of this species (*i.e.* seed set and asexual

reproduction *via* rooting of lateral branches) cannot overcome rates of canopy reduction and mortality (*Lechmere-Oertel et al., 2008*). The environmental buffering effects of *P. afra* increase the amount of soil organic matter (*Lechmere-Oertel et al., 2008*), improving water infiltration (*van Luijk et al., 2013*) and water holding capacity (*Mills & de Wet, 2019*). The canopy of *P. afra* creates the cool shaded microhabitat required for the recruitment of canopy tree species (*Sigwela et al., 2009*; *Wilman et al., 2014*), thus facilitating community assembly processes. The loss of *P. afra* cover triggers a series of feedback loops that steers the ecosystem towards an alternate state with limited ecological functioning. The lack of vegetation cover exposes the soils to erosion which results in the depletion of soil organic carbon (*Cowling & Mills, 2011*; *Lechmere-Oertel et al., 2008*; *Lechmere-Oertel, Kerley & Cowling, 2005*; *Mills & Fey, 2004*), and disrupts hydrological processes such as water infiltration and subsequent retention (*Cowling & Mills, 2011*; *Mills & de Wet, 2019*; *van Luijk et al., 2013*). This leads to further loss of ecological processes and ultimately, significant loss of biodiversity (*Fabricius, Burger & Hockey, 2003*; *Sigwela et al., 2009*).

Active restoration of degraded *P. afra* thicket has been tested in experimental plots established across the biome by the government-funded Subtropical Thicket Restoration Project (STRP). This initiative aimed to generate employment that would transition towards funding support from the global carbon market (*Marais et al., 2009*). The high productivity of succulent thicket and the ease of propagation of *P. afra* (*i.e.* through the planting of unrooted branches into degraded habitat: *Mills & Cowling, 2006*; *van der Vyver, Mills & Cowling, 2021*), make this vegetation ideal for the generation of carbon credits, as it sequesters up to 15.4 t $CO_2$ ha$^{-1}$ yr$^{-1}$ (*Mills & Cowling, 2014*). However, ecosystem-level restoration success and carbon sequestration are measured over decades whereas implementation success and restoration trajectories must be monitored over shorter time frames to enable adaptive management. This monitoring is often limited to costly techniques that are time- and labour-intensive when hundreds to thousands of hectares need to be covered. Fortunately, recent technological advances have provided innovative techniques to overcome this challenge in restoration (*Almeida et al., 2021*; *Chen et al., 2021*; *Wang et al., 2021*).

Increased accessibility of UAVs makes the generation of high-resolution remote sensing data available with relatively little sampling effort (*Colomina & Molina, 2014*). These platforms require little specialist training and can provide accurate data at both spatial and temporal scales. Given the acceleration in availability and volumes of this data, automated approaches to analyzing this data are essential to fully exploit its potential (*Kattenborn et al., 2021*). Convolutional neural networks (CNNs) are particularly well suited to vegetation analysis as they extract features in data (*e.g.*, spatial features in the case of aerial imagery) that best describe the target object (such as leaf and canopy shapes, edges between individuals, and spectral properties of individual species). Training the model to extract the desired features is particularly efficient as the algorithm itself is able to learn what patterns are important based on reference material. This is done in sequential batches, enabling the model to process large amounts of data, facilitating the training of models with ample and diverse data so that they are transferable across sites and lighting conditions. CNNs have thus been used to identify individual plant traits such as growth

form (*Fromm et al., 2019*; *Sylvain, Drolet & Brown, 2019*), phenology (*Hasan et al., 2018*), species (*Fricker et al., 2019*; *Wagner et al., 2020*), and communities (*Kattenborn, Eichel & Fassnacht, 2019*) from aerial imagery.

The growing interest in the global carbon market presents restoration initiatives with novel funding structures (*Galatowitsch, 2009*) that are likely to contribute to the upscaling of Thicket restoration (*Marais et al., 2009*; *Mills et al., 2015*). This will require effective monitoring of change in canopy cover at a range of spatial and temporal scales to inform adaptive management practices. By pairing imagery derived from readily available "out-of-the-box" UAVs and CNN segmentation algorithms, this study successfully classifies canopy cover of reintroduced *P. afra* in experimental thicket restoration plots established between 2008 and 2009 (*Mills et al., 2015*). Thus, we demonstrate that commercially available UAVs and CNN algorithms can provide rapid and accurate estimates of *P. afra* cover at low cost and without expert training. Implementation of this monitoring approach will allow efficient monitoring of changes in vegetation cover and facilitate adaptive management through fine-scaled spatio-temporal monitoring of restoration sites.

## MATERIALS AND METHODS

### Study site and UAV data acquisition

Between 2008 and 2009, approximately 300 experimental restoration plots were established in degraded habitat across the global extent of *P. afra* dominated thicket vegetation (as delineated by *Vlok, Euston-Brown & Cowling, 2003*). Each plot consisted of a 0.25 ha (50 × 50 m) herbivore exclosure that was fenced to a height of 1.2 m. These experimental plots tested a range of *P. afra* planting strategies (briefly described in *van der Vyver, Mills & Cowling, 2021*), but produced highly variable results, with survival ranging between zero and nearly 100% between plots (*van der Vyver et al., 2021*). The complete or partial removal of fencing exposed some experimental plots to herbivory (*van der Vyver, Mills & Cowling, 2021*), and differences in local climatic and soil conditions (*Vlok, Euston-Brown & Cowling, 2003*) may have resulted in differences in the rate of *P. afra* survival and growth, resulting in highly variable cover between these plots (*Duker et al., 2020*). In order to train and test the CNN-segmentation models, aerial imagery was acquired for thirty-two experimental plots that reflect this variability in cover.

For this, RGB imagery was acquired from thirty-two plot-specific flights in 2020–2021 using a DJI Phantom 4 Pro ($n = 12$) and DJI Mavic 2 Pro ($n = 20$). Flights were conducted at different times and dates, resulting in variable image brightness and contrast, and the size and casting of shadows. The flight height was approximately 30 m above ground, with 10 m spacing between photographs, resulting in a Ground Sampling Distance (GSD) of ~0.9 cm/pixel. Flight plans were generated using a custom script in R (V1.4.1717), and FlyLitchi (V4.22, www.flylitchi.com) was used to operate the UAV during flights. Imagery was stitched using Metashape (V1.7.2; Agisoft LLC, Saint Petersburg, Russia). Examples of the images generated are provided in Fig. 1.

Reference data was generated by visual interpretation of the orthoimagery, where *P. afra* crowns were mapped by means of manually drawn polygons (using QGIS 3.24.0). For each orthoimage and plot, the reference data acquisition targeted an area of approximately 12.5

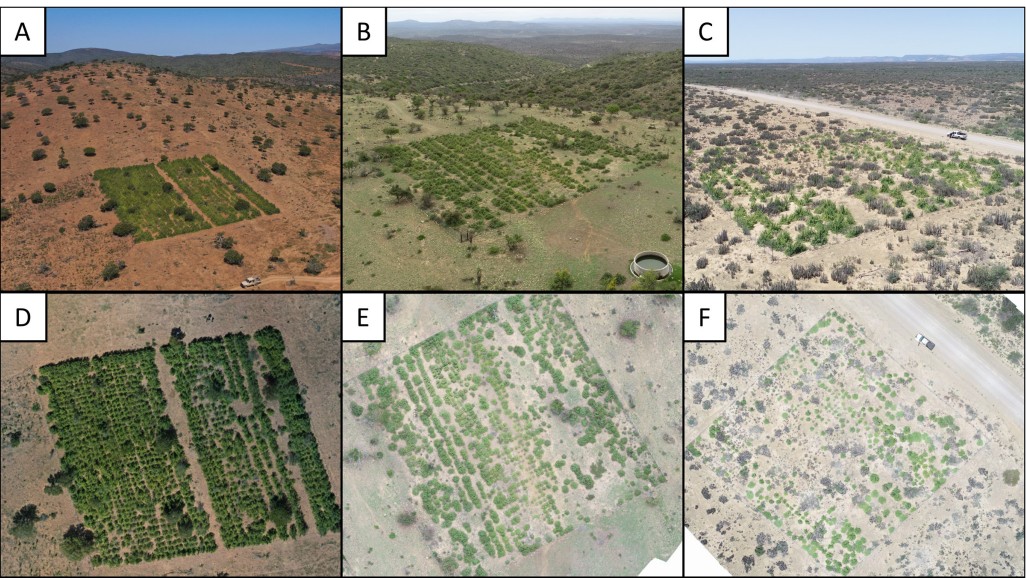

**Figure 1 Photographs of three experimental restoration plots.** (A–C) Experimental restoration plots in context. Note the open woodland (degraded thicket) surrounding the plots in relation to the dense woodland (intact thicket) in the background. (D–F) Arial images of the above restoration plots used for *P. afra* canopy cover classification.

× 12.5 m. After visual interpretation, the shapefiles were converted to a binary mask (presence *vs.* absence of *P. afra*) with a raster resolution corresponding to the respective orthoimage.

## CNN model training and validation

For training the CNN, non-overlapping tile pairs of 128 × 128 pixels were seamlessly cropped from the orthoimages (predictors) and the masks (reference). For this, only the visually interpreted (*P. afra* cover) portion of the reference images was considered (as per previous section). The tile size of 128 × 128 pixels was chosen as at this spatial extent the characteristic features of *P. afra* are still visible and large amounts of non-overlapping tiles are available for model training.

As a segmentation algorithm, we implemented the Unet architecture (*Ronneberger, Fischer & Brox, 2015*), which is known for its efficiency and widely used for segmentation tasks in high resolution remote sensing images (*Kattenborn et al., 2021*). The Unet architecture is composed of an encoder and a decoder part, which are linked with skip connection (Fig. 2). Both the encoder and decoder parts contain pooling operations, which reduce the spatial resolution of the feature maps. In the encoder part, the model extracts the image features for detecting *P. afra* at multiple spatial scales. The skip-connections transfer the activation maps of each spatial scale to the decoder part, which, hence, enables segmentation of the crown dimensions at the original spatial resolution of the input imagery. Here, we used encoder and decoder parts composed of four convolutional blocks, where each block consists of three convolutional layers followed by a batch normalization and max pooling operation. In the consecutive blocks, the convolutional layers had a depth of 512, 256, 128 and 64 layers for the encoder part and equally but in reverse order for the

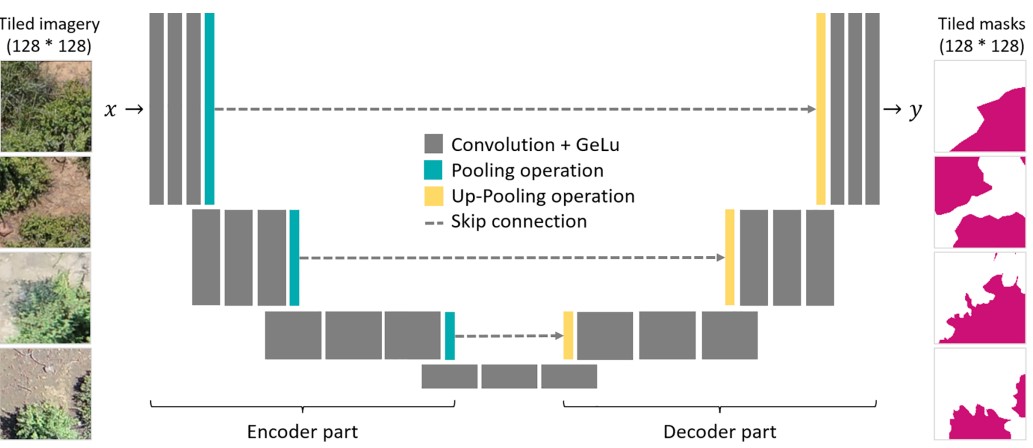

**Figure 2 A schematic illustration of the Unet architecture and the tiled pairs of UAV imagery (left) and corresponding binary segmentation masks (right).**

decoder part. As activation functions, we used Gaussian Error Linear Units (GELU). Similar setups have been successfully applied in previous studies (*Kattenborn, Eichel & Fassnacht, 2019*; *Schiefer et al., 2020*).

To avoid optimistic model evaluation by spatially autocorrelated training and validation data (*Ploton et al., 2020*; *Kattenborn et al., 2022*), we randomly split all available data on a plot basis, where a portion of plots was used for model training ($n = 24$) and the remainder used for model testing ($n = 8$). The training data was again split in training (7/8) and validation data (1/8), whereas the validation data was used to monitor the training process. The models were trained in 200 epochs and the final model used for further analysis was selected based on the lowest loss on the validation data (see S1 for training *vs.* loss-rate curve). As a loss function, we used binary cross-entropy:

$$Loss = \frac{1}{N} \sum_{i=1}^{N} y_i - \log \widehat{y}_i + (1 - y_i) - \log(1 - \widehat{y}_i),$$

where $N$ are the number of model outputs, $\widehat{y}_i$ is the model output and $y_i$ the target value.

Training the model took ~450 min using an NVIDIA A6000. The final model performance for the tiles that were included in the training, validation and testing was reported using the F1-score (also known as dice coefficient). Additionally, we performed a t-test to assess if F1-scores differed significantly between imagery obtained with the DJI Phantom 4 Pro and the DJI Mavic 2 Pro.

The final model was used to predict *P. afra* crowns in all orthoimagery. The prediction was performed using a moving window approach, in which individual tiles of the same size as used for model training (128 × 128 pixels) were seamlessly cropped from the orthoimagery. The final model was then applied on these tiles and the predictions were stored as a prediction raster containing the class probability (0 = absence & 1 = presence of *P. afra*). To reduce edge effects (potential mispredictions at the border of tiles), we applied this procedure two times, where the locations of extracting the tiles were shifted by 50% of

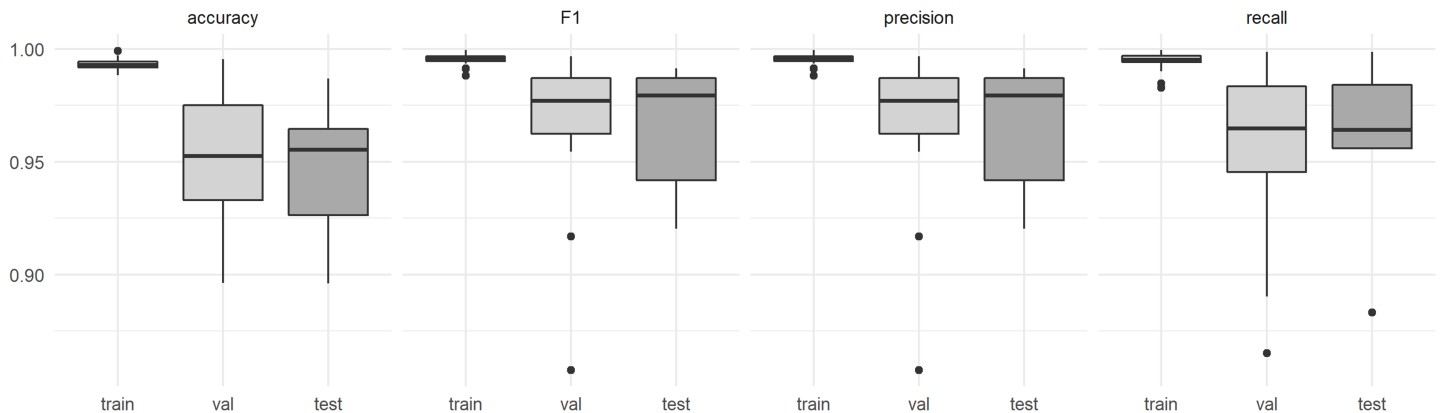

**Figure 3** Model performance estimates derived from the (train) training data, (val) the validation data and (test) data derived from entirely independent plots and respective UAV acquisitions.

the tile size (64 pixels). The average of two resulting prediction rasters were calculated and a threshold of (0.5) was applied to produce a binary classification output.

## RESULTS

The model performance in terms of F1-score was 0.995 for the training data, 0.969 for the validation data and 0.966 for the test data obtained from the entirely independent plots. The F1-score for the individual plots in the test dataset was at least 0.920 (Fig. 3). Visual interpretation revealed that lower accuracies resulted from misinterpretations of the reference data. No significant difference in F1-scores was detected between predictions obtained for the DJI Phantom 4 Pro and the DJI Mavic 2 Pro (t = 1.5283, df = 24.84, *p*-value = 0.1391, Fig. 3). The recall (mean = 0.962) and precision (mean = 0.965) metrics derived from the test dataset indicate that no systematic over- or underprediction of *P. afra* occurred (Fig. 3).

## DISCUSSION

The application of CNN machine learning models proved suitable for the classification and quantification of *P. afra* cover in the highly variable RGB aerial imagery, generated from commercially available 'out of the box' UAV models (Fig. 4). This approach to species mapping has been widely applied for the classification of tree species in forest communities (*Fricker et al., 2019*; *Natesan, Armenakis & Vepakomma, 2019*; *Pinheiro et al., 2020*; *Schiefer et al., 2020*) and appears to be well suited to thicket vegetation that exhibits a similar closed canopy structure (*Vlok, Euston-Brown & Cowling, 2003*). Furthermore, the method presented here proved to be transferable across different UAV models, sites and lighting conditions, with no apparent loss of performance (Fig. 3).

Where multiple UAV flights are required for data collection, it is common practice to limit flights to the same time of day and on clear days so as to minimize the potential effects of solar angle and radiance on model performance (see examples in *Abdulridha, Batuman & Ampatzidis, 2019*; *Adak et al., 2021*; *Eskandari et al., 2020*; *Guo et al., 2020*; *Lopatin et al., 2019*). This was not the case here. UAV data acquisition was not restricted to specific illumination conditions, no image corrections or cross-calibrations were
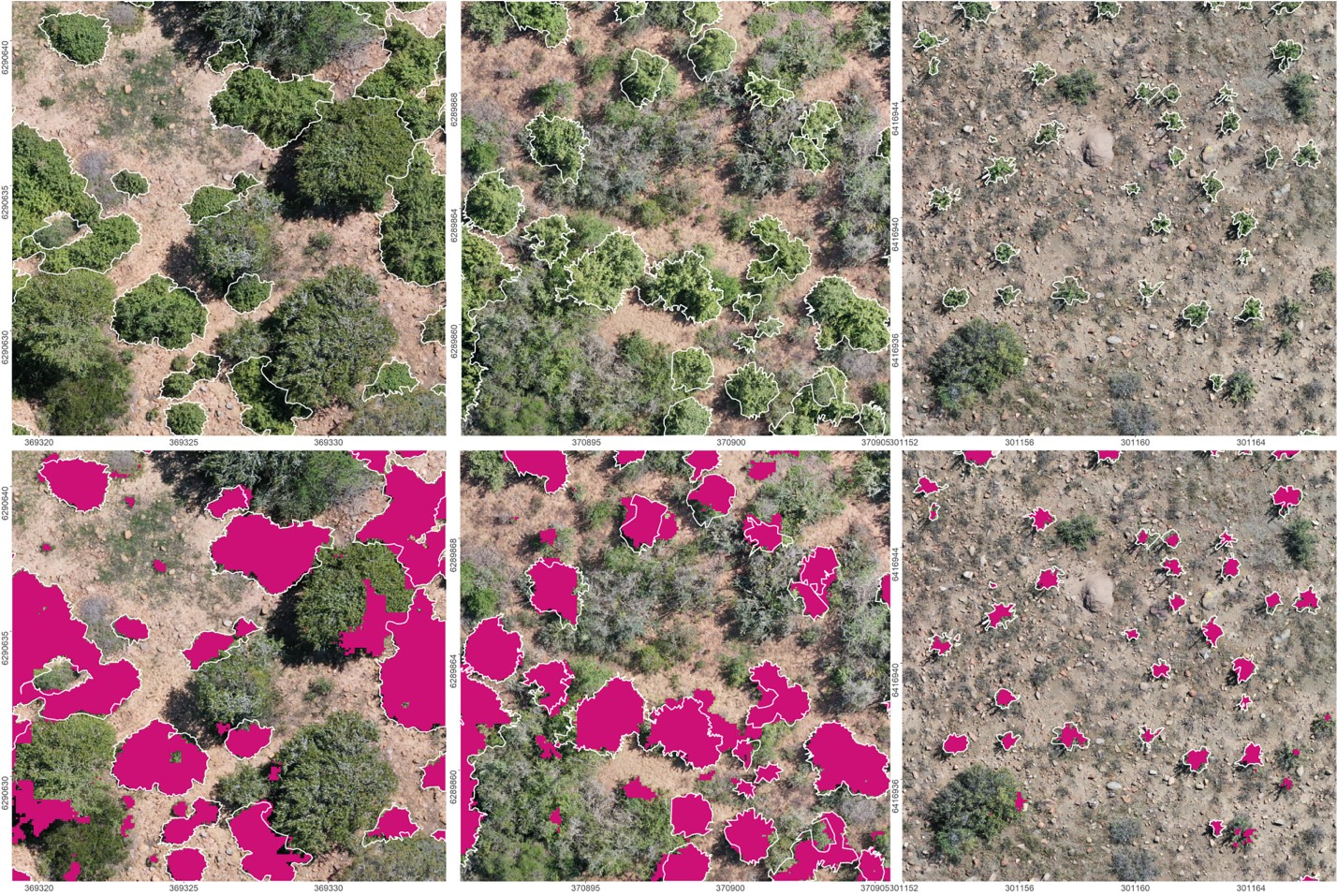

**Figure 4 Prediction results of the final CNN model on the orthoimagery.** Top: The orthoimagery overlaid by the reference polygons (white). Bottom: Orthoimagery overlaid with reference polygons (white) and segmentation results (purple). EPSG: 32735.

conducted, and the model was trained using images sourced under a range of conditions and using different models of UAVs. Despite this, model performance was comparable to other studies using CNN-based segmentation approaches (see studies reviewed in *Kattenborn et al., 2021*), indicating that the model learned robust representations of *P. afra* under a range of the above mentioned factors. Note, that the transferability of the models was successfully tested with entirely independent image acquisitions, which may even include acquisition and site conditions that were not explicitly included in model training (cf. *Kattenborn et al., 2022*). Thus, the model presented here could be applied to quantify *P. afra* cover in restoration sites across the Albany Subtropical Thicket biome in South Africa.

The ease of use and transferability of aerial image classification models presents new opportunities for defining and tracking restoration targets across a range of spatial and temporal scales. *Loewensteiner et al. (2021)* demonstrate the importance of temporal scale in defining restoration targets, by applying CNN models to classify woody cover in a

Savanna ecosystem over a 66 year time period. Woody cover was found to vary over time, and thus restoration targets for the system should fall within the range of woody cover detected and are not required to reflect the current vegetation structure of the reference ecosystem. Similarly, the model presented here could be applied to images of intact thicket ecosystems to better describe restoration targets and planting densities, as the current practice aims to generate a dense closed canopy by reintroducing *P. afra* at high densities (1–2 m spacing). Additionally, it may be possible to detect the return of ecosystem functioning using aerial imagery. This may include measures of structural complexity, indicative of biodiversity returns (*Camarretta et al., 2020*), or regeneration dynamics (*e.g.*, measures of target species cover as presented here for *P. afra*, and for pioneer forest species in *Wagner et al., 2020*) and seedling recruitment (*Buters, Belton & Cross, 2019*; *Fromm et al., 2019*).

The data presented here demonstrates that thicket restoration initiatives could efficiently analyze aerial imagery collected using different UAV models for temporal monitoring of restoration trajectories, and inform adaptive management practices (*Camarretta et al., 2020*). This, currently, presents a major challenge in thicket restoration as field-based monitoring often takes place in remote areas where *P. afra* has coalesced to form dense and impenetrable stands. Repeat aerial imagery can be collected for restoration sites with no increase in sampling effort over time, and with little effort or training. The data generated can potentially be sent to a centralized repository for analysis, bridging the science-practice gap (*Dickens & Suding, 2013*) and promoting further collaboration within the Albany Subtropical Thicket restoration community (*Mills et al., 2015*). This will provide managers with estimates of plant density (for example, by using blob detection, which separates individuals in the CNN classification: *Kattenborn et al., 2021*) and assist in tracking changes in plant cover to ensure interventions can be made if plant cover is lost due to disturbance (*e.g.*, frost: *Duker et al., 2015a*, *2015b*, or herbivory: *van der Vyver et al., 2021*). In such cases, actions can be informed by the scientific community and implemented by managers and landowners to remediate the processes threatening the restoration initiative.

The approach presented here demonstrates the potential of monitoring *P. afra* cover at local scales with great efficiency and flexibility, in rather inaccessible sites. However, the spatial extent of UAV acquisitions with sufficient resolution is constrained by battery life—for example, with the UAVs used in this study, imaging over approximately 15 ha requires a fully charged battery when flying at a height of 60 m and multiple UAV acquisitions would be required to evaluate *P. afra* cover for larger areas. However, other UAV designs, such as fixed-wing, offer even better area coverage, and it is very likely that image acquisition over large areas will be routinely achievable (*Meneses et al., 2018*).

An alternative approach for monitoring larger areas could also include the integration of satellite imagery, where UAV-based estimates on *P. afra* cover are used to train machine learning models with data from Earth Observation satellite data, such as from the Sentinel or Landsat missions (see *Kattenborn, Eichel & Fassnacht, 2019*; *Fraser, Van der Sluijs & Hall, 2017*; *Fraser, Pouliot & van der Sluijs, 2022*; *Naidoo et al., 2021*). Although such satellite imagery features considerably lower spatial resolution, their high temporal and

spectral resolution can suffice to differentiate plant species and hence enable them to extrapolate cover from the local (UAV) to regional scales.

Another constraint of the proposed approach, from the practitioner's perspective, may be access to sufficient computing capabilities. High-resolution remote sensing data can result in large data volumes and, although once trained, CNN models are quite efficient, they are ideally applied with GPUs to process large datasets (*Flood, Watson & Collett, 2019*; *Brandt et al., 2020*). Cloud computing can provide a possible means of overcoming these computational loads for processing large datasets (see *Kattenborn et al., 2021* for some cloud platforms with GPU support), making the classification of large spatial areas feasible for a greater number of practitioners. Cloud computing in combination with integrating remote sensing data may thus prove invaluable for the upscaling of Thicket restoration in South Africa, with an estimated 1.2 million ha of degraded ecosystems having some restoration potential (*Lloyd, van den Berg & Palmer, 2002*).

While we have presented the first CNN model relevant to Albany Subtropical Thicket, and that could be applied to monitoring restoration at scale, we do not harness the full capabilities of machine learning in this study. Here, we used the well-known Unet algorithm (*Ronneberger, Fischer & Brox, 2015*), while CNN-based segmentation algorithms are steadily advancing (*Minaee et al., 2021*). UAV and sensor technologies also continue to advance, with improved image resolution and the spectral range of drone imagery becoming ever cheaper, which will make detection of finer-scale patterns possible without having to decrease flight altitudes. Additionally, the three-dimensional mapping of vegetation cover using LiDAR technologies provides opportunities to estimate plant biomass without labor intensive fieldwork (*Shendryk et al., 2020*; *ten Harkel, Bartholomeus & Kooistra, 2019*). This may aid in calculating carbon sequestration in restoration sites to assist in carbon credit verification and generation for sale on the global carbon market. *Harris, Bolus & Reeler (2021)* present a satellite-based example of this, reporting a significant correlation between above-ground carbon stocks calculated from remotely sensed *P. afra*-dominated vegetation canopies and field measures. In addition, increased image resolution will likely provide better insights into *P. afra* recruitment dynamics, and further developing the CNN model to classify multiple species (as per *Fricker et al., 2019*; *Kattenborn, Eichel & Fassnacht, 2019*) can provide insights into biodiversity return with minimal sampling effort. This is a laborious task to complete using manual field measures, last undertaken by *van der Vyver et al. (2013)*, requiring multiple days of field surveys to cover relatively small areas—this can now be covered within a matter of hours using UAVs by a non-specialist. Additionally, field measures are more prone to sampling biases, whereas the repeated use of a CNN model will provide consistent and repeatable results. Thus, it is evident that the work presented here should inspire the future application of UAV imagery to the ecology and management of Albany Subtropical Thicket vegetation.

## CONCLUSIONS

Recent advancements in machine learning and remote sensing technologies have provided unprecedented ease of processing aerial imagery. This can potentially transform monitoring of ecosystem restoration practices, shifting protocols from slow and expensive

field measures to more efficient remote sensing approaches. Here we demonstrated the utility of standard 'out of the box' UAV data coupled with CNN models to classify and quantify *P. afra* cover in thicket restoration plots. The models were transferable across imagery collected with different UAV models and under varying lighting conditions. The integration of this model in the monitoring of thicket restoration will aid in the planned upscaling of Albany Subtropical Thicket restoration and generate valuable temporal data for evaluating restoration trajectories and demographic processes. This will promote collaborative efforts between scientists and practitioners, strengthening the restoration community. Importantly, the integration of this monitoring approach does not require any technical knowledge of the CNN model, or special skill sets to fly commercially available UAVs, and can thus improve efficiency while reducing the cost of sampling efforts required for accurate monitoring of restoration at landscape scales.

## ACKNOWLEDGEMENTS

The authors would like to thank Dr Marius L. van der Vyver for his assistance in acquiring UAV imagery and Anize Tempel and Kristen Hunt who assisted in manual classification of images for model training.

### Funding

This work was supported by the National Research Fund of South Africa (Grant No. 119379) and the Nelson Mandela Universities' postdoctoral research fellow grant program. The collection of UAV imagery was funded by the Natural Resource Management programme of the South African Department of Forestry, Fisheries and the Environment (Project No. E1406). The funders had no role in study design, data collection and analysis, decision to publish, or preparation of the manuscript.

### Grant Disclosures

The following grant information was disclosed by the authors:
National Research Fund of South Africa: 119379.
Nelson Mandela Universities' Postdoctoral Research Fellow Grant Program.
Natural Resource Management Programme of the South African Department of Forestry.
Fisheries and the Environment: E1406.

### Competing Interests

Alastair J. Potts is an Academic Editor for PeerJ.

### Author Contributions

- Nicholas C. Galuszynski conceived and designed the experiments, prepared figures and/or tables, authored or reviewed drafts of the article, and approved the final draft.
- Robbert Duker conceived and designed the experiments, performed the experiments, prepared figures and/or tables, authored or reviewed drafts of the article, and approved the final draft.
- Alastair J. Potts conceived and designed the experiments, performed the experiments, analyzed the data, prepared figures and/or tables, authored or reviewed drafts of the article, and approved the final draft.
- Teja Kattenborn conceived and designed the experiments, performed the experiments, analyzed the data, prepared figures and/or tables, authored or reviewed drafts of the article, and approved the final draft.

## Data Availability

All input data and outputs (model objects, predictions, statistics) are available at Zenodo: Galuszynski, Nicholas, Potts, Alastrair, Duker, Robert, & Kattenborn, Teja. (2022). Spekboom UAV imagery and reference data (2022-09-02) [Data set]. Zenodo. https://doi.org/10.5281/zenodo.7044728.

They are also available at GitHub with all the code: https://github.com/tejakattenborn/unet_spekboom.

## Supplemental Information

Supplemental information for this article can be found online at http://dx.doi.org/10.7717/peerj.14219#supplemental-information.

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
