# Peer review of "Automated mapping of Portulacaria afra canopies for restoration monitoring with convolutional neural networks and heterogeneous unmanned aerial vehicle imagery"

_PeerJ, doi:10.7717/peerj.14219_

## Round 0.1 · original submission · Minor Revisions

Overall, the discussion part is weak. The Discussion should summarize the manuscript's main finding(s) in the context of the broader scientific literature and address any study limitations or results that conflict with other published work.

Reviewer 1 ·

Basic reporting

The text is written clearly. The authors say what they want to do.

Experimental design

The image are collected from the DJI drone. It was collected at different times and dates and this should be fine.

Validity of the findings

I think the findings are plausible and make sense.

Additional comments

I think the paper is fine. I would like to suggest to the authors if I may.
1. Please provide a reason why do you decide to use an image size of 128 x 128 pixels?
2. The author used binary cross-entropy as an objective function. I suggest showing the equation and giving a few details.
3. I suggest that the authors should provide the modified U-net architecture image. it will help the reader understand the U-net architecture.

Reviewer 2 ·

Basic reporting

In this article, the authors proposed an approach for deforestation monitoring by applying a CNN-based segmentation model to accurately classify the canopy cover of Portulacaria afra.

The training datasets are acquired through the UAV imaging platform. Authors are applying a cutting-edge approach to solving a real challenge. However, The article must be written in unclear and ambiguous English and must be written in a form that can attract readers.

Requesting the authors to review the article properly in terms of clear understandable writing.

Experimental design

The experimental set-up design and its analysis are found inadequate throughout the article. Such as,

- Should provide sample training images.
- The reason behind using the Unet model
- Choosing 100 epochs, probably explain the training vs. loss-rate curve.
- Validations results are also important to explain the experimental setup.
- Which type of hyperparameters were tuned and why tuned?
- Proper analysis of segmentation results by applying TP, TN, FP, FN, and accuracy.
- What is the technical shortcoming in the dataset since acquired through UAV?
- The output results need to upload along with the dataset.

Additionally, UAV capturing and its applications are very dependent on image resolution and lighting conditions after eliminating noise. Authors should provide a rigorous justification of how they overcome the limitations of UAVs such as windy, noisy, and bad weather (challenging) conditions.

Validity of the findings

Many works have already been done using the machine learning approach in deforestation. In the article, the authors should provide varieties of dataset images to show how challenging it is to make the proposed approach robust. Also, the claims need to be validated.

Only an F1 score is not enough to justify it.

·

Basic reporting

No Comment.

Experimental design

No Comment.

Validity of the findings

No Comment.

Additional comments

None.

---

## Round 0.2 · accepted · Accept

I congratulate the authors for the effort put into this paper! The manuscript is significantly improved; therefore, I recommend accepting it in its current form!

Reviewer 1 ·

Basic reporting

no comment

Experimental design

no comment

Validity of the findings

no comment